# Genomic and Epigenomic Characterization of Tumor Organoid Models

**DOI:** 10.3390/cancers14174090

**Published:** 2022-08-24

**Authors:** Chehyun Nam, Benjamin Ziman, Megha Sheth, Hua Zhao, De-Chen Lin

**Affiliations:** Center for Craniofacial Molecular Biology, Herman Ostrow School of Dentistry, and Norris Comprehensive Cancer Center, University of Southern California, Los Angeles, CA 90033, USA

**Keywords:** biomarker, genomics, epigenomics, organoid, patient-derived organoid, pharmacogenomics

## Abstract

**Simple Summary:**

The patient-derived organoid (PDO) model is a versatile and dynamic tool for investigating individual genomic and epigenomic properties of cancer. PDOs preserve key (epi-)genomic features and maintain physiological and pathological characteristics, resembling patient tumor specimens. Coupled with the next-generation sequencing technology, PDOs can be used to accurately map (epi-)genomic alterations of “living” cancer specimens. Additionally, organoid modeling of matched normal and malignant tissues from the same patient serves as an adequate and valid platform to interrogate cancer-specific features. Because of these advantages, research on PDOs is rapidly growing for the investigation of genotype–phenotype association and precision oncology.

**Abstract:**

Tumor organoid modeling has been recognized as a state-of-the-art system for in vitro research on cancer biology and precision oncology. Organoid culture technologies offer distinctive advantages, including faithful maintenance of physiological and pathological characteristics of human disease, self-organization into three-dimensional multicellular structures, and preservation of genomic and epigenomic landscapes of the originating tumor. These features effectively position organoid modeling between traditional cell line cultures in two dimensions and in vivo animal models as a valid, versatile, and robust system for cancer research. Here, we review recent advances in genomic and epigenomic characterization of tumor organoids and the novel findings obtained, highlight significant progressions achieved in organoid modeling of gene–drug interactions and genotype–phenotype associations, and offer perspectives on future opportunities for organoid modeling in basic and clinical cancer research.

## 1. Introduction

Cancer is a disease of the genome and epigenome. Over the last 15 years, we have witnessed the revolution of research on cancer genomics and epigenomics, propelled by chemical, imaging, and computational breakthroughs in sequencing technologies. Large-scale, often consortium-led tumor sequencing projects have not only led to the full annotation of the genomic landscape of most human cancers, but also established new cancer taxonomies and subtypes, often complementing traditional histology-guided classifications. These genomic and epigenomic sequencing efforts are the driving forces of precision oncology. Indeed, routine genomic profiling with the intent of improving diagnosis and prognosis has already been realized in several cancer types, as exemplified in breast cancer (e.g., HER2 amplification), non-small cell lung cancer (e.g., EGFR mutation, ALK fusion), and melanoma (e.g., BRAF mutation). 

Despite these tumor sequencing breakthroughs being immensely powerful and informative, significant challenges and obstacles exist, hindering broad and efficient implementation of genome-based precision medicine, not least because in many cases, there is an apparent disconnection between genomic aberrations and biological consequences. In fact, comprehensive cancer sequencing studies have generated many outstanding and important questions: how to efficiently and rigorously distinguish driver from passenger mutations, and relatedly, how to systematically prioritize the functional significance of the large number of genomic aberrations from tumor sequencing, in addition to how to associate molecular-based taxonomy and classification with cancer biology and therapeutic response. Among the multitude of genomic alterations for which targeted therapies are predicted, which of the druggable targets offer truly meaningful efficacy? What are the founding genomic and epigenomic alterations during the earliest steps of neoplastic evolution and transformation? Addressing these questions carries enormous scientific and translational value, ultimately reshaping genome-based cancer medicine. Clearly, however, tumor sequencing by itself is insufficient to tackle these key challenges, and additional tools and model systems are required. In particular, a dynamic and valid human disease model that allows for functional characterization of genomic and epigenomic alterations in a faithful and robust system would be enormously beneficial. 

In this regard, in vitro organoid modeling of human cancer serves as a promising bridge between static, descriptive tumor sequencing and functional annotation of the cancer genome and epigenome. Patient-derived tumor organoids (PDOs) preserve key genomic and epigenomic features of the modeled tumor, maintain physiological and pathological characteristics of the original sample, and actively divide, self-renew, and self-organize into 3D multicellular structures. Matched nonmalignant and cancerous organoids can be derived from the same patient, providing adequate and valid controls for biological comparison. Additionally, generating and maintaining organoids is considerably more efficient and cost-effective than patient-derived xenograft (PDX) models. Moreover, it is now possible to culture organoids long-term from a number of cancer types, which is imperative for resource sharing and data reproducibility. Together, these unique advantages make organoid culturing a valid, versatile, and robust model to investigate the cancer genome and epigenome. Indeed, there is growing interest and enthusiasm in applying this technology to study the functional significance of genomic and epigenomic alterations, model spatial genomic heterogeneity and longitudinal genome evolution, associate therapeutic response with driver mutations, and investigate the earliest changes during neoplastic evolution and transformation. For example, a variety of epigenomic assays can be employed using PDOs, including whole-genome bisulfite sequencing (WGBS) to measure global DNA methylation levels, ATAC-seq to assess the chromatin accessibility, and ChIP-seq to determine interactions between DNA and protein (Figure 1).

## 2. Genomic Sequencing of Patient-Derived Tumor Organoids

Whole-genome, -exome, or targeted panel sequencing have been applied to PDOs from cancers of the ovary, stomach, pancreas, esophagus, bladder, and colon, among others (Table 1). Comprehensive genomic profiling of PDOs has also been extended to rare tumor types, such as pediatric kidney tumors [1]. These data reveal that in general, early passaged PDOs faithfully maintain genomic mutations and copy number changes from their tumors of origin. For example, in a genomic study of pancreatic ductal adenocarcinoma (PDAC) PDOs [2], 82.49% to 99.96% of the mutations found in the primary tumor samples were successfully detected in the organoid culture. Moreover, DNA mutational signatures (“footprint” of genome mutational processes) also show fairly high concordance between PDOs and originating tumor specimens [3]. Nevertheless, during long-term culture, PDOs often undergo genomic evolution and drift. For example, under 2-month in vitro culture, liver cancer PDOs retained ~92% of original somatic mutations; this preservation rate decreased to 80% over four months in culture. Colon cancer PDOs derived from samples with mismatch-repair deficiency developed numerous new mutations after a 6-month culture, while the mismatch-repair proficient counterparts had a much more stable genome during long-term culture [4,5]. In addition, PDOs from certain cancer types (such as HNSCC) [6] may be more susceptible to genetic drift in vitro than others. Multiple mechanisms can drive this genomic evolution over time, including tumor-intrinsic factors (e.g., genomic instability, mismatch-repair deficiency) and tumor-extrinsic factors (e.g., sub-clonal selection by certain growth factors and inhibitors in the culture media). In those cancer types with inherently low tumor purity (e.g., PDAC, ovarian cancer), more somatic mutations were sometimes found in the PDOs because of the high neoplastic cellularity of the organoids. This increased detection sensitivity in PDOs is even more so with respect to copy number alterations [2], supporting PDO as an alternative, more sensitive means of identifying genomic abbreviations in tumors with low purities.

## 3. PDO Pharmacogenomics Linking Driver Mutations to Therapeutic Response

Interrogating gene–drug association and linking drug sensitivity to genomic sequencing data represent perhaps one of the most important applications of organoid modeling in cancer. One of the earliest studies [8] of this kind generated and sequenced 20 PDOs from colon cancer patients, followed by screening using a library of small-molecule inhibitors. Among these PDOs, those with KRAS hotspot mutation were resistant to treatment with cetuximab (an anti-EGFR inhibitor), highly consistent with findings from clinical trials. A recent large-scale genomic study of over 100 primary and metastatic breast PDOs [7] correlated “BRCAness” (BRCA1- or BRCA2-associated mutational signatures) [25] with the sensitivity to poly (ADP-ribose) polymerase (PARP) inhibition, recapitulating the established genetic interaction between BRCA1/2 and PARP in homologous recombination repair [26]. In HNSCC [6], PDO lines unresponsive to cetuximab treatment often had mutations downstream of the EGFR pathway, including *PIK3CA*, *KRAS*, *HRAS*, and *BRAF*. This finding has potential implications for genetic testing guidance on patient selection for cetuximab therapy. In prostate cancer, PDOs [23] with androgen receptor gene amplification were exquisitely and specifically sensitive to enzalutamide (androgen receptor antagonist), whereas those harboring both PTEN loss and PIK3R1 mutation were sensitive to PI3K pathway inhibitors (Everolimus and BKM-120). In PDAC [2], Afatinib (an inhibitor targeting ERBBs) had higher activity toward PDOs harboring *ERBB2* amplification. In addition, bladder cancer PDOs [3] with FGFR3 gain-of-function mutations showed specific sensitivity to pharmacological inhibitors against MEK and ERK.

In addition to the investigations of PDOs established directly from clinical cancer specimens, gene–drug association studies have also been applied to genetically engineered organoids derived from normal tissues that have undergone malignant transformation induced by oncogenic alterations. A notable strength of such organoid models is the clean, well-controlled, and genetically defined background. For example, CRISPR/Cas9-mediated ARID1A and TP53 dual knockout in primary human gastric organoids induced mucinous differentiation and tumorigenicity. High-throughput chemical screening identified that ARID1A-deficient gastric organoids were uniquely vulnerable to inhibition of BIRC5/survivin [18]. Our group recently [27] generated a novel model based on wild-type and TP53/CDKN2A dual-knockout (DKO) human normal gastroesophageal junction (GEJ)-derived organoids edited by using CRISPR/Cas9. DKO organoids grew faster, became larger, and exhibited de novo intestinal, metaplastic, and dysplastic morphology. Moreover, DKO organoids consistently underwent xenograft growth in vivo. Notably, platelet-activating factor receptor (PTAFR) was uniquely upregulated upon TP53/CDKN2A dual knockout, which rendered DKO organoids sensitive to a pharmacologic inhibitor of PTAFR. These forward oncogenic transformation approaches have also been used in human embryonic stem cell (hESC)-derived organoids to characterize drug responses associated with defined driver mutations. In a retinoblastoma organoid model derived from genetically engineered hESCs with a biallelic mutagenesis of RB1, the tyrosine kinase SYK was significantly upregulated. Consequently, these retinoblastoma organoids were highly sensitive to SYK inhibition [24]. 

The approach of genetic-engineering-based tumorigenesis has also been extended to normal organoids from murine tissues, coupled with gene–drug interaction screens. Genome-edited, malignant-transformed murine organoids carry an additional advantage: they serve as syngeneic lines which can be investigated in immunocompetent tumor models. A recent study genetically engineered mouse organoids derived from fallopian tube epithelium, and developed them into high-grade serous ovarian cancers [15]. Three major subtypes with patient-informed mutational combinations were created by CRISPR/Cas9 genome editing: *Trp53**^−/−^; Ccne1^OE^; Akt2^OE^; Kras^OE^*, *Trp53**^−/−^; Brca1**^−/−^; Myc^OE^*, and *Trp53**^−/−^; Pten**^−/−^; Nf1**^−/−^*, representing homologous recombination-proficient, -deficient, and unclassified subtypes, respectively. Shallow whole-genome sequencing identified different copy number abbreviations and aneuploidies between these subtypes of cancer organoids. In drug response assays, *Trp53**^−/−^; Brca1**^−/−^; Myc^OE^* organoids expectedly showed heightened sensitivity to PARP inhibitors. On the other hand, gemcitabine killed *Trp53**^−/−^; Ccne1^OE^; Akt2^OE^; Kras^OE^* organoids more effectively than the other models, consistent with the previous finding linking CCNE1 overexpression to DNA replication stress [15,16]. Moreover, in a syngeneic model, tumors formed by *Trp53**^−/−^; Ccne1^OE^; Akt2^OE^; Kras^OE^* organoids were immunologically “hotter”, showing higher infiltration with T cells, macrophages, and mDCs than the other two genotypes. These features rendered them more sensitive to immune blockade therapies, including anti-PD-L1 and anti-CTLA4 antibodies. These findings strongly support cancer organoids as a valid and robust model system to study gene–drug interactions and correlate therapeutic vulnerabilities with genomic alterations.

## 4. Establishing Genotype–Phenotype Correlations by PDOs

The versatile and amenable nature of organoid modeling also facilitates the interrogation or placement of driver mutations in signaling cascades, including upstream signals and downstream activities in native environments. A number of such studies have been published. For example, in the 20 whole-exome sequenced colon PDOs [8], one organoid culture with RNF43 mutations was uniquely sensitive to the inhibition of the Wnt pathway, consistent with the finding that RNF43 negatively regulates the Wnt pathway, by removing the Wnt receptor FZ [28,29]. Another study generated and profiled 39 lines of PDAC PDOs using both whole-exome sequenced and comparative genomic hybridization microarray analyses [22]. Functional studies demonstrated that driver mutations dictated the requirements for the corresponding niche and growth factors. Indeed, mutations of KRAS, SMAD4, and TGFBR2 correlated with the reliance of organoids on EGF, Noggin removal/BMP4 treatment, and A83-01 removal/TGF-b1 treatment, respectively. TP53 mutations/in-del alterations were associated with the sensitivity to Nutlin3 treatment (an MDM2 inhibitor). Intriguingly, Wnt/R-spondin dependency was largely unrelated to somatic mutations, activating the Wnt signaling in these PDOs, which led the authors to further investigate the exogenous origin of Wnt ligands.

In a study of gastric cancer PDOs [19], 37 lines of organoids were analyzed by whole-exome sequencing and copy number analyses. Interestingly, PDOs containing both TP53 and CDH1 mutations grew in an R-spondin-independent manner, connecting the dual mutation of TP53/CDH1 (enriched in the diffuse subtype of gastric cancer) to the Wnt pathway. Indeed, genetic engineering to knockout both TP53 and CDH1 in normal gastric organoids was sufficient to generate the growth independence [19] of R-spondin, strongly supporting the gene–pathway association observed in the PDOs. This study also reported that gastric cancer PDOs with ERBB2/3 gene amplifications were able to grow in the absence of EGF and were sensitive to the treatment of a pan-ERBB receptor kinase inhibitor. These results together underscore the utility of PDO culture as a model system to interrogate genotype–phenotype correlations.

## 5. Organoid Modeling of Spatial Genomic Heterogeneity and Longitudinal Genome Evolution 

One of the most comprehensive genomic intratumor heterogeneity studies was from 78 PDOs [10] derived from multiple tumor regions from three colon cancer patients. As anticipated, truncal (early) driver mutations (e.g., those affecting APC, KRAS) were shared among organoids from the same tumor regions. However, in some cases, organoids derived from spatially close regions still had substantial differences in overall mutation burden and mutational signatures. When tested against a panel of chemotherapeutic drugs and targeted agents, different PDOs from the same tumor displayed striking differences in drug response. Some of the differences were associated with certain driver mutations. For example, truncating mutations in RNF43 [29] were correlated with sensitivity to the blockade of the Wnt pathway. Such spatial intratumor heterogeneity of drug sensitivity has also been documented in a PDO study of PDAC [2], wherein four different PDOs were generated from the liver, diaphragmatic metastases, and ascites from the same patient. These genomically sequenced PDOs exhibited different sensitivities to 5-fluorouracil (5-FU). However, the genomic underpinnings of the varied responses remain obscure. Intratumor genomic heterogeneity was also observed in a whole-genome sequencing study of 36 ovarian cancer PDOs from 23 patients [14]. These PDOs displayed both inter- and intra-patient heterogeneity of response to chemotherapy and targeted therapies, which were partially explained by genetic alterations. For example, PDOs with copy number changes in ATP7A and ATP7B genes exhibited differential responses to chemotherapies, consistent with prior findings of the effect of copper efflux pumps on chemotherapy sensitivity [30,31]. These organoid heterogeneity results identify both convergence of earlier truncal events and sub-clonal genomic divergence. Even organoids derived from the same tumor region sometimes show highly variable mutational signatures and drug responses.

Longitudinal tumor genomic diversification and evolution has also been characterized using organoid modeling. As an example, longitudinal sequencing of serially passaged bladder PDOs [3] revealed that truncal mutations were often retained, while sub-clonal (late) mutations were prone to be gained or lost. CRISPR/Cas9 knockout of MLH1, a central DNA repair factor often mutated in colon cancer, promoted mutagenesis in normal colonic organoids [11] over time. Analogously, also in normal colonic organoids, dual deletion of APC and TP53 triggered abnormal chromosome segregation and aneuploidy [13]. Organoid modeling of tumor genomic evolution has also been investigated in the context of drug treatment. For example, a 35-day in vitro treatment of 5-FU in colon organoids induced a specific mutational pattern, characterized by T > G mutations in a CTT trinucleotide context. Importantly, this pattern was also detected in vivo from human cancer samples treated with 5-FU [9]. These studies highlight the value and validity of the organoid modeling system for longitudinal analyses of cancer genomic evolution.

## 6. Transcriptomic Analysis of Tumor Organoids Identifies Novel Cancer Subtypes

In addition to genomic profiling, epigenomic characterization has also been extensively applied to organoid modeling to study the cancer epigenome, spanning the transcriptome, DNA methylome, and chromatin states. A notable advantage of epigenomic profiling of tumor organoids is the direct measurement of cancer cells without stromal contamination (fibroblasts, endothelial cells, immune cells, etc.). These works have established novel cancer subtypes, identified new drug–epigenome associations, and revealed the functional significance of epigenetic changes in cancer. 

Not surprisingly, gene expression profiling of PDOs almost always separates normal and tumor organoids into distinct clusters: examples can be found in colon [8], breast [7], pancreatic [2,22], and gastric [19] cancers. Among tumor organoids, sub-clusterings are often observed from transcriptomic analyses. In an RNA-seq analysis of 44 PDAC PDOs [2], classic and basal-like subtype signatures were identified, consistent with previous findings from PDAC patient data [32]. In addition, the classic subtype of PDO cultures offers a valuable experimental resource, since very few cell line models of this PDAC subtype are currently available. In another characterization of 39 PDOs from PDAC patients [22], three functional subtypes were established based on their dependencies on niche factors Wnt and R-spondin. Interestingly, clustering analyses of the matched transcriptomes revealed a near-linear correlation between gene expression clusters and Wnt niche subtypes. Specifically, the linear trajectory began from normal organoids, progressed through Wnt-dependent PDOs, and finally reached the Wnt/R-spondin-independent subtype, indicating serial transition and acquisition of gene expression networks contributing to the Wnt niche independence. In addition, the Wnt/R-spondin-independent subtype exhibited high levels of gene signatures reflecting the basal type of PDAC. Moreover, GATA6 was identified as a key regulator of this novel Wnt/R-spondin-independent phenotype. Interestingly, an analogous subtype of Wnt/R-spondin-independent PDOs was also identified in gastric cancer [19]. RNA-seq analysis showed that these gastric PDOs had a unique gene expression pattern, with specific upregulation of a number of X-chromosome-linked cancer-testis genes. Notably, the gene signature derived from Wnt/R-spondin-independent PDOs was significantly correlated with inferior outcomes in gastric cancer patients. In bladder cancer, RNA-seq analyses clustered PDO [3] cultures into either basal or luminal subtypes using two independent molecular classifiers, consistent with findings from primary bladder tumors.

In addition to PDOs, normal organoids transformed by defined oncogenic drivers also inform transcriptomic changes associated with different genotypes, as exemplified in de novo generation of three subtypes of ovarian cancers using murine fallopian tube epithelial organoids [15]. Indeed, HR-proficient, -deficient, and unclassified organoid models had distinct transcriptomes, which were associated with different secretomes regulating the tumor microenvironment in various manners. 

## 7. DNA Methylome and Chromatin Accessibility Profiling of Tumor Organoids 

Besides transcriptomic profiling, other epigenomic data, such as DNA methylation and chromatin modifications, were also able to cluster PDOs into different subgroups with biological underpinnings. For example, Wnt/R-spondin-independent PDAC PDOs exhibited a unique DNA methylome compared with their Wnt/R-spondin-dependent counterparts [22]. GATA6, the upstream regulator of the Wnt/R-spondin-independent group, was epigenetically silenced by DNA hypermethylation at its gene promoter. In gastric cancer [19], genome-wide DNA methylation analysis identified a subgroup consistent with the CpG island methylator phenotype (CIMP [33]) established in patient samples [34]. CIMP^+^ gastric PDOs were enriched in the microsatellite instability subtype with MLH1 hypermethylation, in agreement with previous findings from primary tumor sequencing data [34]. The CIMP^+^ subtype has also been confirmed in a subset of colon cancer PDOs [10]. In this study, principal component analysis of the methylome and transcriptome was performed to investigate both inter- and intra-tumor epigenomic heterogeneity, showing that although clones from the same patient largely clustered together, sub-clones distributed separately. As a notable example, TP53 wild-type and mutant clones from the same patient were distinguished by genome-wide methylation and transcription profiles. 

Organoid modeling has also facilitated investigations of the interplay and cooperation between genomic and epigenomic cancer drivers, which is particularly challenging to model. In a study of long-term in vitro culture (12–14 months) of colon organoids [12], DNA methylation array profiling showed that promoter DNA hypermethylation was spontaneously acquired in an aging-like manner. These epigenetic changes silenced key genes of the Wnt pathway, leading to a progenitor-like cellular state, which was susceptible to neoplastic transformation by the driver mutation Braf^V600E^. In comparison, short-term cultured “young” organoids lacking such promoter DNA hypermethylation were much more resistant to Braf^V600E^-induced transformation. Indeed, it took Braf^V600E^ 5 months to transform young organoids, in stark comparison to merely 2 weeks for their older counterparts. Importantly, CRISPR/Cas9-mediated knockout of key Wnt pathway genes targeted by DNA hypermethylation was able to phenocopy the aging-like, spontaneous epigenetic silencing. These results not only reveal the complex interaction and interplay between DNA hypermethylation and Braf^V600E^ during early tumorigenesis, but also highlight organoid modeling as a superb system for investigating the functional relationship between genomic and epigenomic alterations in cancer.

In addition to the DNA methylome, chromatin accessibility landscapes of tumor organoids have also been characterized. For example, a recent work performed the assay for transposase-accessible chromatin with sequencing (ATAC-Seq) on 41 PDOs from pancreatic cancer patients [21]. These PDOs were derived from various histological classifications, including PDAC, intraductal papillary mucinous neoplasms, acinar cell carcinoma, and pancreatic neuroendocrine neoplasms, allowing for the association between differential chromatin accessibility and tumor histopathology. Indeed, motif enrichment analysis based on ATAC-seq peaks facilitated the identification of transcription factors potentially operational in each subtype. As anticipated, in neuroendocrine PDOs, there was a significant enrichment of neuroendocrine cell-type-specific transcription factors, such as ASCL1, NEUROD1, and NKX2-5. On the other hand, HNF4A and TP63 were enriched in the classical- and basal-like subtypes of PDAC, respectively, consistent with previous observations from primary patient data [35,36]. 

## 8. Identification of Drug–Epigenome Interactions by PDOs

In-depth, multi-layered epigenetic characterizations of PDOs have enabled the establishment of drug–epigenome interactions. A high-throughput drug screening of the pancreatic cancer PDOs [21] revealed chromatin accessibility signatures associated with sensitivity to either cytotoxic chemotherapeutics or targeted inhibitors. For example, an ATAC-seq peak assigned to the NCOR2 gene was correlated with cellular sensitivity to Go6976, an inhibitor of the JAK/STAT3 pathway, congruent with the finding that NCOR2 modulates activity of the JAK/STAT3 pathway [37]. Another ATAC-seq peak predicted to regulate BAG3 transcription was found to be significantly associated with cellular resistance to 5-FU and paclitaxel treatment, in line with the results that BAG3 renders cancer cells unresponsive to 5-FU and paclitaxel [38,39]. In another study of PDAC PDOs [2], a drug sensitivity expression signature was computed by correlating PDO transcriptomic profiles with drug efficacy values. Notably, the drug sensitivity expression signature predicted treatment response in PDAC patients from different cohorts, suggesting that these signatures derived from PDO culture may provide potential biomarker value in predicting therapeutic outcome in PDAC patients.

## 9. Single-Cell RNA-seq of Organoids to Identify Cancer Cell-of-Origin

Organoid epigenomics has also facilitated the identification of cancer cell-of-origin, a vital subject for not only delineating the mechanistic basis of tumorigenesis, but also developing potential avenues for early cancer detection and prevention. However, tumor-initiating cells of many cancer types still remain obscure and sometimes debatable. For example, although a significant amount of attention has been placed on the originating cells of retinoblastoma, prior research has come to different conclusions. Various cell types, such as horizontal cells, Müller glial precursor cells, human cone precursors, and retinal progenitors have all been proposed as cells responsible for cancer initiation [40,41,42,43,44,45,46,47]. These disparities may result from either the discordance between the murine and human disease characteristics, or the lack of a robust and relevant human cancer model. To address these weaknesses, single-cell genomics has recently been applied to early tumorigenesis organoid models to map cancer cell-of-origin. 

For instance, regarding retinoblastoma, Liu and colleagues [24] genomically edited human embryonic stem cells (hESCs) with either homozygous RB1 loss-of-function mutation or deletion, and then differentiated these mutant cells into retinoblastoma organoids. Bioinformatic “lineage-tracing” was conducted based on single-cell RNA-seq coupled with pseudo-time trajectory construction and RNA velocity analyses. These computational deconvolution approaches located ARR3^+^ maturing cone precursors at the branching point of the phylogenetic tree of cancer initiation, sequentially followed by retinoma-like and retinoblastoma cells, suggesting ARR3^+^ maturing cone precursors as tumor-initiating cells of retinoblastoma. Intriguingly, a similar single-cell RNA-seq study [48] of retinoblastoma organoid models instead proposed proliferating cone precursors (RXRγ^+^Ki67^+^) as the cancer cell-of-origin. The divergence possibly stemmed from the patient-specific genetic background, which influences the initiation and development of retinoblastoma. Specifically, in addition to normal hESC cells, the latter work generated and characterized retinoblastoma organoids from a patient-specific induced pluripotent line.

Single-cell RNA-seq has also been utilized to analyze metaplastic and dysplastic organoid lines derived from KRAS-driven murine stomach tissues [49], which represent early precursor stages of gastric cancer. Reduced dimensional projection of single-cell transcriptomes completely separated metaplastic and dysplastic organoids, suggesting distinct gene expression networks between these two premalignant cellular states. Metaplastic organoid cells were largely clustered together, indicating low cellular heterogeneity. In comparison, dysplastic organoids were separated into a minor, metaplastic-like subset and a major, dysplastic-specific population, with each cluster expressing signature genes associated with premalignant biological processes. These single-cell genomic and bioinformatic analyses of premalignant organoids have important implications in investigating the earliest stages of tumorigenesis. 

## 10. Conclusions and Future Perspectives

Because of the distinctive biology and unique advantages, organoid modeling technologies have not only revolutionized the genomic and epigenomic studies of human cancer, but also transformed our understanding of the fundamentals of a “living tumor”. Consequently, organoid culturing is rapidly becoming a mainstream methodology for both basic and translational cancer research. Nonetheless, in spite of many successful and promising studies thus far, considerable challenges and significant bottlenecks remain to be resolved. These challenges are both technical and scientific. At the technical level, success rates of organoid culturing vary substantially across different cancer types, as well as across different samples from the same cancer type. Indeed, certain tumor types are extremely difficult to model by organoid culture. Other notable technical caveats include: (i) the initial outgrowth of nonmalignant epithelial cells, which can only be identified and confirmed by genomic sequencing; (ii) complex combinations of niche factors and growth factors in the culture media, which not only interfere with cancer biology but also vary across different laboratories, affecting reproducibility and data sharing; (iii) relatedly, culture protocols for generation and maintenance of PDOs are not standardized, hampering its translational and clinical development. 

At the scientific level, despite early passaged PDO cultures generally preserving and retaining the genomic makeup of the originating tumors, the magnitude of culture-associated genomic evolution and drift is unclear, with ensuing biological significance and consequences undetermined. Another apparent limitation of current organoid methods is the lack of integration and incorporation of native stromal components of the tumor microenvironment. These stromal cells regulate extracellular matrix, vascularity, angiogenesis, and anti-tumor immunity, therefore playing a central role in modulating drug response and tumor aggressiveness. Addressing this particular obstacle, organoid co-culture systems are emerging and promising results have been achieved in the incorporation of fibroblasts [50,51] and T cells [52] This development of organoid co-culture systems will be particularly helpful for the in vitro study of cancer immunology and immunotherapy.

Moving forward, many opportunities also exist for further developing organoid culture to better model tumor biology and therapeutic response. One example is the opportunity to integrate cancer-related environmental factors, such as oncogenic viruses and bacteria, into organoid modeling. In fact, initial explorations have already yielded interesting results in Helicobacter-pylori-associated gastric cancer [53] and Salmonella-associated gallbladder cancer [54]. In addition, lifestyle cancer risk factors, such as tobacco, alcohol, and dietary factors, are envisioned to be modeled by organoid culture. Moreover, the current scale of most PDO “living biobanks” is still rather limited. The establishment of large-scale PDO biorepositories of diverse tumor types and genotypes will greatly aid the identification of novel genomic and epigenomic drivers, systematic screening of drug–gene interactions, and mechanistic research of cancer biology.

## Figures and Tables

**Figure 1 cancers-14-04090-f001:**
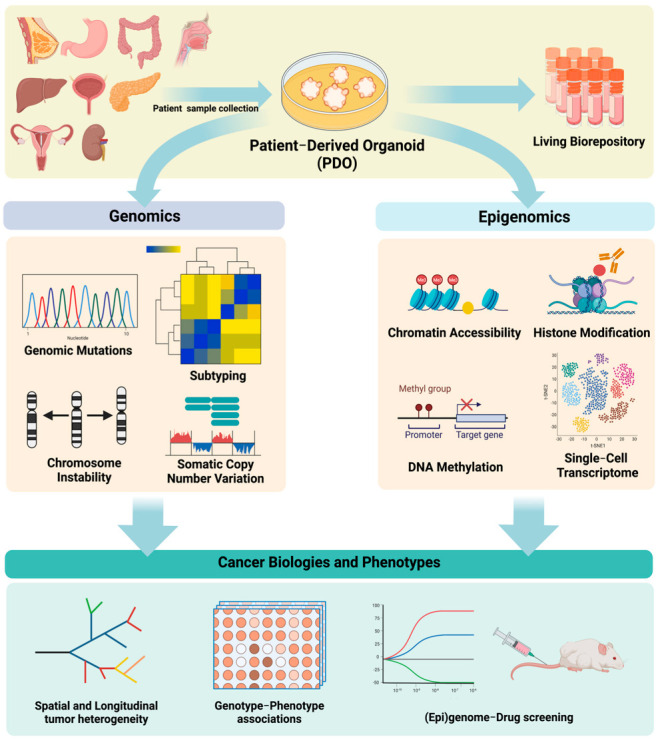
Genomic and epigenomic analyses of patient−derived tumor organoids and its application in cancer research.

**Table 1 cancers-14-04090-t001:** Overview of genomic and epigenomic analyses of PDOs.

Cancer Type	Ref.	Species	Method	Sample No.	Major Subtypes /Classifications	Correlation with Drugs
Breast cancer	[7]	Human	WGS	101	PAM50, SCMGENE/SCMOD1, ER−/HER2−, HER+	Afatinib, Gefitinib, Pictilisib, GDC-0068, AZD8055, Everolimus, and Tamoxifen
			RNA-seq	22		
Bladder cancer	[3]	Human	WES	24	Basal, luminal	26 chemicals
			RNA-seq	42		
Colorectal cancer	[5]	Human	WES	43	Adenoma, serrated, MSS, MSI, and NEC	A83-01, SB202190
			Target-seq	19		
	[8]	Human	WES	41	Hypermutated, non-hypermutated	17-AAG, 5-FU, Cetuximab, GDC0941, Gemcitabine, MK-2206, Nutlin-3a, NVP-BEZ235, and SCH772984
			RNA-seq	108		
	[9]	Human	WGS	3	-	5-FU
	[10]	Human	WGS	73	BRAF/ACVR2A, APC/TP53, KRAS/APC	Doxorubicin, SN38, 5-FU, Afatinib, Nutilin3a
			RNA-seq	76		
			HM450K	70		
	[11]	Human	WGS	30	WT and mutant MLH-1	Y-27632
	[12]	Mouse	Bisulfite Pyro-seq	8	BRAF^V600E^	IWP-2, IWR-1-endo, and CCT031374
	[13]	Human	WGS	6	KRAS^G12D^, APC^KO^, P53^KO^, SMAD4^KO^	Gefitinib, Noggin, A83-01. and SB202190
Epithelial ovarian cancer	[14]	Human	WGS	36	HR-proficient, TP53, BRAF, KRAS, NRAS, XIAP, and CDKN2a	Alpelisib, Adavosertib, Afatinib, AZD8055, Carboplatin, Gemcitabine, MK-2206, Niraparib, Olaparib, Plitaxel, Pictilisib, Rucaparib, Vemurafenib, Flavopiridol, Cobimetinib
	[15]	Mouse	WGS	12	Trp53^−/−^; Ccne1^OE^; Akt2^OE^; Kras^OE^, Trp53^−/−^; Brca1^−/−^; Myc^OE^, and Trp53^−/−^; Pten^−/−^; Nf1^−/−^	Rucaprib, Niraparib, Olaparib, Gemcitabine, Doxorubicin, Paclitaxel, Carboplatin, Seliciclib, PHA767491, BAY1895344, Chloroquine
			RNA-seq	12		
	[16]	Human	WES	34	BRCA1/2	Carboplatin, Olaparib, Prexasertib, and VE-822
ESCC	[17]	Mouse	WES	58	-	ADAR1 inhibitor
			RNA-seq	14		
			ChIP-seq	44		
			ATAC-seq	8		
Gastric cancer	[18]	Human	WGS	3	MSI- and EBV-type	YM-155
			RNA-seq	6		
	[19]	Human	WES	46	CIMP+, CIMP−, and normal/normal like	Y-27632, EGFR/ErbB-2/ErbB-4 inhibitor, Nutlin-3, Crizotinib, and C59
			GEM	62		
			EPIC array	51		
	[20]	Mouse	WES	6	WT and mutant TP53	AZD7762, Prexasertib
			RNA-seq	20		
HNSCC	[6]	Human	WES	24	-	Cetuximab, Cisplatin, Alpelisib, Vemurafenib, Everolimus, Nutlin-3 and AZD4547
			RNA-seq	16		
Pediatric kidney cancer	[1]	Human	WGS	59	Wilms tumor, malignant rhabdoid tumor, renal cell carcinoma, congenital mesoblastic nephromas	Vincristine, Actinomycin D, Doxorubicin, Etoposide, Panobinostat, Romidepsin, PD-0325901, Idasanutlin
			RNA-seq	51		
			EPIC array	45		
Pancreatic cancer	[21]	Human	WGS	35	Classical-like, basal-like, classical-progenitor, Glycomet	283 epigenetic-related chemicals, 5 chemotherapeutic drugs
			RNA-seq	87		
			ATAC-seq	44		
	[2]	Human	WGS	22	Classic, basal-like, or C1, C2	Afatinib, Gemcitabine, Paclitaxel, SN-38, 5-FU, and Oxaliplatin
			WES	69		
			RNA-seq	49		
	[22]	Human	WES	48	WNT−, WNT+, WRi	A83-01, SB202190, Nutlin-3, and C59
			GEM	18		
			EPIC array	25		
Prostate cancer	[23]	Human	WES	7	TMPRSS2-ERG fusion, SPOP mut, SPINK1 overexpression, and CDH1 Loss	Enzalutamide, Everolimus, and BKM-120
			RNA-seq	7		
RB	[24]	Human	RNA-seq	8	-	R406, Bay61-3606, and Rapamycin
			WGBS	8		

ESCC, esophageal squamous cell carcinoma; GEM, gene expression microarray; HNSCC, head and neck squamous cell carcinomas; No, number; Ref, Reference; RB, retinoblastoma; WT, wild type; WRi, Wnt, and R-spondin-independent; CIMP, CpG island methylator phenotype; WGS, whole-genome sequencing; WES, whole-exome sequencing; WGBS, whole-genome bisulfate sequencing; HM450K, Infinium HumanMethylation450 BeadChip; EPIC array, Infinium MethylationEPIC Kit.

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
