# Peer review of "Genomic and Epigenomic Characterization of Tumor Organoid Models"

_cancers, 2022, doi:10.3390/cancers14174090_

Round 1
Reviewer 1 Report
In this manuscript, the authors reviewed the advances in (epi-)genomic characterization of tumor organoid models. In this decade, the field of 3D organoid culture has been developing fast. Various cancer and normal tissue 3D organoid models have been reported and such models are good tools for drug screening and personalized medicine. This review give a summary on the studies that tumor organoids can provide (epi-)genomic information to indicate tumor heterogeneity, genotype-phenotype association and drug screening, etc. The manuscript is well written and clearly structured. The figure is very helpful to highlight the main focus of the manuscript. The table is informative. I recommend it is accepted.
Author Response
We are very appreciative of your evaluation of our work
Reviewer 2 Report
The review manuscript submitted by Nam et al. entitled "Genomic and epigenomic characterization of tumor organoid models" to Cancers Journal aims to provide a deep review on the ultimate studies of the field. The authors reviewed in deep the current advantages and disavantages of using tumor organoid models in genomic and epigenomic characterization of tumors and corresponding progression gained in organoid modeling of gene-drug interactions and genotype-phenotype interactions. Finally, the authors review the current perspectives on future chances in using organoid modellin in basic and clinical cancer research and treatment. The references used in the manuscript are well-choosen and the text is very well-written. Hovewer, the following points should be improve before the acceptance to publication:
- In the introduction section, authors should detail in deep (descriminate better) which epigenomic assays are available to employ in organoids models;
- Please reformat the table 1 (e.g. the lines for each cancer type)
- Please reformat the #1
Author Response
1) In the introduction section, authors should detail in deep (descriminate better) which epigenomic assays are available to employ in organoids models;
In the introduction section, we have detailed which epigenomic assays are available to employ in organoids models.
2) Please reformat the table 1 (e.g. the lines for each cancer type)
We have reformatted the Table-1. In particular, the lines are now without format issues.
3) Please reformat the #1
We adjusted the margin of the reference #1.
please check the attached file
